# Benefits and Conflicts: A Systematic Review of Dog Park Design and Management Strategies

**DOI:** 10.3390/ani12172251

**Published:** 2022-08-31

**Authors:** Shuolei Chen, Zhuoran Wu, Ole Russell Sleipness, Hao Wang

**Affiliations:** 1College of Landscape Architecture, Nanjing Forestry University, Nanjing 210037, China; 2Department of Landscape Architecture and Environmental Planning, Utah State University, Logan, UT 84321, USA

**Keywords:** dog ownership, dog park, dog walking, health benefits, off-leash area, physical activity

## Abstract

**Simple Summary:**

Dog parks contribute physical and social benefits for both canines and their owners, especially during and since the COVID-19 pandemic. However, dogs in public places can create various conflicts. Growing numbers of scholars have explored strategies for effective park design and management. This systematic study synthesizes and analyze the benefits, conflicts, and strategies for the design and management of dog parks according to the PRISMA guidelines. Based on the summary of conflicts between canines, humans, and their environment, we present design and management guidance for dog parks to effectively mitigate these conflicts while enhancing the benefits of off-leash areas. While this study promotes a sustainable and healthy coexistence of canines and residents of built environments through appropriate design and management strategies, several research and practice gaps have been identified from the results, such as the dearth of experimental evidence and limitations of the physical benefits of dog parks. These research gaps provide opportunities for experts to address in future.

**Abstract:**

Dog ownership and dog walking brings various health benefits for urban dwellers, especially since the COVID-19 pandemic, but trigger a number of controversies. Dog parks have become increasingly significant public resources in the pandemic to support these benefits while facing intense conflicts. To develop effective dog parks in urban settings, growing numbers of scholars have provided insights into the design and management strategies for addressing the benefits and conflicts. The objective of this study is to synthesize and analyze various aspects of dog park design and management and to assess identified strategies for enhancing their benefits while mitigating their drawbacks. Following the PRISMA guidelines, a systematic study was conducted to synthesize the benefits, conflicts, and management strategies of dog parks, supported by Citespace. Benefits and conflicts in dog park design and management have been synthesized and organized according to their frequency of presence and the statistical results. We analyzed and assessed existing design and management strategies. Through this systematic study, we discovered the need obtain o po experimental evidence on effective dog park design and management to enhance their benefits while mitigating their sources of conflict and limitations in the intensity of park visitors’ physical activity in off-leash areas. Guidelines for the design and management strategies for effective dog parks were made to enhance their benefits while alleviating conflicts in the future development of sustainable dog parks that promote healthy relationships between canines and residents in urban built environments.

## 1. Introduction

### 1.1. Dog Ownership and the Impacts

Dog-ownership accounts for a significant proportion of households across countries [1]. A high proportion of dog ownership provides various benefits, including increased physical activity [2,3], social and mental health benefits [4,5], reduced cardiovascular risk [6], and all-cause mortality [7], as identified by both experimental studies and systematic reviews. During and since the COVID-19 pandemic and the associated lockdowns, restrictions, dog ownership, and attachment have been found to be related to fewer mental and physical health issues [8,9,10,11,12]. Additional studies documented the health benefits of owning dogs for various groups, including children [13,14], the elderly [15], and populations with disabilities [16,17]. However, in the meantime, the presence of domestic dogs in urban areas is subject to various environmental and social conflicts arising from the presence of dog feces [18] and conflicts between dog owners and non-dog owners in public open spaces [19,20]. 

### 1.2. Growing Demands for Dog Parks

While many countries and cities retain strict regulations for dogs in public spaces, the prevalence of dog ownership and both its positive and negative impacts amidst increasing urban densities, highlight the need for safe and controllable environments for dogs and their owners. Given this situation, the development of dog parks therefore has emerged a solution. A dog park is defined as a designed off-leash area offering opportunities for people and their dogs to socialize and exercise legally [21,22]. Most dog parks are built within larger urban parks although some created as stand-alone parks [23,24]. Dog parks are a feasible option for satisfying the physical and social needs of dogs and their owners, while providing separation for non-dog owners who may be offended by dogs. However, minimizing the issues commonly found with dog parks inevitably relies on effective design strategies [23]. The COVID-19 pandemic has resulted in a significant increase in dog park visitation [25]. The growing need for dog parks has shown a need for research that analyzes the benefits and issues of dog parks for the benefit of scholars, designers, park managers, and policymakers. 

### 1.3. Aspects of Dog Parks

Many scholars have asserted that as built environmental resources, dog parks tend to strengthen a myriad of positive impacts of dog ownership [26,27,28]. The primary benefits of dog parks can be generalized into physical and social dimensions. Many studies have acknowledged that a nearby dog park can encourage physical activity, through dog walking and play, which consequently contributes to human and canine physical health [22,23,29,30,31]. Moreover, dog parks provide a space for dogs and their owners to meet and become acquainted with each other, which enhances social interactions for both individuals and their dogs [26,32]. Socializing in dog parks can result in greater positive feelings towards the neighborhood, enhancing the sense of community and social capital [18,22,23,24,32,33,34]. Other related benefits of dog parks include reduced aggressiveness of dogs, resulting in better controlled dogs [21], and reduced criminal activity [23]. These benefits should be advocated through effective design strategies for dog parks.

Although dog parks provide benefits for both individuals and their dogs, considerable opposition in the design process should not be overlooked. Some issues caused by domestic dogs are often aggravated in dog parks because of the concentrated gathering of dogs. Some studies indicated negative impacts of dog waste on plants, causing soil erosion, unpleasant odors, and the transmission of diseases [23,35]. Canine aggression including dog fights and bites can be a severe phenomenon in dog parks and may result in injuries and controversies. However, it remains questionable if the environmental and health issues related to humans and their dogs can be addressed in the design of dog parks [36].

The design of the built environment can either facilitate or hinder activities such as dog walking [2]. In order to strengthen related activities within dog parks, an increasing number of design guidelines/strategies have been developed for researchers and practitioners. With the advancement of research, most recent studies indicate design strategies should consider the benefits and problems of dog parks beforehand [21]. Systematic studies have also suggested that the primary goal of dog park design should be to enhance their benefits while reduce their conflicts in place [23].

### 1.4. Research Objective

Dog parks are constructed to provide opportunities for individuals and their dogs to achieve health benefits while mitigating the conflicts between people, animals, and the environment. Design strategies for dog parks should target strengthening these benefits and while mitigating their problems. Even though emerging research suggests that dog park design strategies should be formulated in terms of their benefits and conflicts, no study has investigated whether the existing design strategies match the increased benefits and conflicts of dog parks, or assess if the design of dog parks can support the provision of health benefits while relieving these conflicts. To address these research gaps, this systematic study first synthesizes and analyzes the existing pros and cons and design/management strategies of dog parks. Based on the analysis, we provide recommendations for future planners, researchers, and decision-makers to optimize the design and research processes for dog parks. To achieve the research objective, the following detailed research questions were explored in this study:(a)What are the existing benefits of dog parks?(b)What are the conflicts that have happened in the dog parks?(c)What are the design strategies for dog parks?(d)What are the management strategies for dog parks?(e)How is it possible to endorse the benefits and minimizing the conflicts while determining the design and management strategies of dog parks?

## 2. Method

To answer the research questions, a systematic study was conducted following PRISMA (Preferred Reporting Items for Systematic reviews and Meta-Analyses) [37], using the support of Citespace to investigate the knowledge structure of canine-related relevant studies. 

### 2.1. Search Criteria and Strategy

Inclusion criteria of this study are English-language, peer-reviewed journal articles and full-text academic dissertations and theses [38], in which dog parks are mentioned or the thematic focus. More specifically, our review focused on articles pertaining to: (1) aspects of domestic dogs and dog-related activities in urban open space, including both the benefits and problems; (2) design and/or management strategies of dog parks/off-leash areas of urban open space. Articles that did not focus on the settings of dog parks or urban off-leash areas were excluded from this review. 

First, Citespace was employed to determine the knowledge structure of relevant fields, by exploring the development and importance of the dog park research. Second, the in-depth systematic review was conducted following PRISMA. In order to fully cover the relevant concepts of dog parks, the search key words included: “dog park” OR “off-leash area” OR “dog walking” OR “dog ownership” OR “canine”. An online search with these keywords was conducted in Google Scholar, Scopus, PubMed, and Web of Science. Results from the keyword search were scanned for their titles and abstracts to determine the full-text articles for analyses. Additional literature that aligned with the search criteria, detected from the reference lists of the full-text articles, were also included for subsequent screening. 

### 2.2. Data Extraction and Analysis

According to PRISMA, relevant content from the selected articles was extracted and analyzed. In addition to the benefits, problems, and design/management strategies of dog parks, we also explored research objectives, methods, and connections between the pros and cons and the design and management strategies. In order to address the core research questions of this study of whether/how the existing design/management strategies of dog parks correspond to the identified benefits and problems, we analyzed the strategies, problems, and benefits of dog parks, as well as the logic underpinning the development of design/management strategies. 

## 3. Results

### 3.1. Citespace Analyses

For the topics of “dog walking” or “dog ownership”, and “benefit”, Citespace conducted with the WOS displayed the time span of retrieval from the year of 1990 to 2022. A total of 1276 journal articles were obtained. The number of studies has grown steadily since 2000, which indicated the topic is worthy of in-depth discussion and research. 

Figure 1 illustrates the knowledge map of the discipline distribution structure of the 1276 articles. As the largest circles are related to veterinary science, the majority of the total searches has been performed in the disciplines of Veterinary Science. With the emergence of circles for other keywords over time, the disciplines of dog ownership and dog walking research have gradually become distributed broadly, from the veterinary disciplines focusing on pet dogs to people-centered social sciences, the environment and ecology, health, and other fields. 

The keyword time zone map (Figure 2) shows the distribution of the keywords, their frequencies, and relationships over time from 1990 to 2022, with the time slice set to every year. Prior to the year 2000, based on the keywords of walking, physical activity, and exercise, we can see that research was focused on the physiology, behavior, and movement. After 2000, the associations between people and pet dogs attracted more attention as research objects. Since 2005, the occurrence of similar keywords began to increase, and dog ownership in the search terms was put forward for the first time. Keywords such as health, human health, and public health have gradually become greater areas of focus. Researchers also began to be concerned about whether dog walking and dog ownership brought other effects besides health, such as risk factors, impacts, and perception. In addition, factors affecting dog walking also attracted research attention, including impacts on the environment, as reflected in the keywords of built environment, park, and other place-based keywords. 

### 3.2. Systematic Study Following PRISMA

The Citespace analyses uncovered impacts on dog walking and health from the perspective of physical environments. Most recent research into the canine disciplines has started to switch the focus to related environments, such as dog parks.

The subsequent systematic study was conducted following PRISMA. Figure 3 illustrates the flow of the literature identification, screening, and inclusion, which yields a total of 55 articles of interest, of which 46 were peer-reviewed journal articles and 9 were dissertation/thesis [36,39,40,41,42,43,44,45,46]. Most of these articles were conducted in the global west, especially in the USA, Australia, and Europe. There were 16 articles proposing dog park design and/or management strategies without discussion of their pros and cons, and 13 articles focusing on the benefits and/or conflicts of dog parks. Around half of these articles (26 out of 55) explored both the pros and cons and the strategies of dog parks, but only nine of them formulated design and/or management strategies according to the benefits and conflicts. Dog park benefits, conflicts, and design and management strategies for the 55 articles are summarized in Table 1. 

As some studies addressed multiple aspects, including benefits, conflicts, or design and management strategies of dog parks (Table 1), we synthesized the information in the following figures (Figure 4, Figure 5, Figure 6 and Figure 7) according to their frequencies in the identified studies.

In Figure 4, the most reported benefits brought by dog parks were identified as improving the physical and social health of dogs and their owners. Some other benefits often mentioned by scholars included building a sense of community and enhancing social cohesion, public safety, and community engagement. Individual scholars indicated that the existence of dog parks in the community can increase property values [22], bring vibrancy to the community [57], and enhance emotional attachment of dog walkers [43]. Additional benefits of dog parks, including mental/psychological health benefits, are related to social benefits, such as promoting human socialization and the enhancement of social cohesion and community engagement.

Hygiene problems related to the dog waste is a serious issue in dog parks, as identified by most studies. Figure 5 also showed that incompatible uses between dog owners and non-dog owners, aggressive dogss, and the lack of regulation of dogs received additional attention among large numbers of researchers. While physical health benefits are the most identified dog park benefits for both park visitors and canines, several studies indicated that dog park visitors may have a limited intensity of physical activity. In addition to the negative impacts on the environments indicated by more than one study, such as damage plant community and soil degradation and erosion, Booth [63] was also concerned that the presence of off-leash dogs may influence wildlife in parks.

Design strategies for dog parks include the consideration of their location, size, adjacent park facilities, amenities, and esthetics (Figure 6). Improvements in accessibility and amenities received the most attention among the proposed design strategies, such as increasing park access and the provision of garbage bins for dog waste. Several studies indicated the placement of signage for direction, adequate seating for dog owners, and monitoring programs or equipment for governing off-leash areas. Numerous studies stated that vegetation and plantings also need to be carefully considered in dog parks. Some other design strategies, such as linear-based path design [3], safety amenities, and natural reserves [64], although only discussed by individual studies, were consistent with the common strategies for dealing with identified conflicts, including lack of physical activity and hygienic issues.

In Figure 7, most research has suggested that strengthening public engagement in the decision-making process for dog park construction/management can address the conflicts between canines and humans, as well as between dog owners and non-dog owning park users. A self-enforcement policy that motivates dog-walkers to manage their dog’s waste is important in the off-leash areas. Numerous researchers have raised the issue of environmental impacts caused by the canines, and this should be core to the management process of dog parks. Some other management strategies covered the necessity of having animal control officer presence, policies and penalties for noncompliance [36], and even a banned list of chronic offenders [46] in dog parks. To deal with the conflicts in dog parks, the strategies of periodic monitoring of soil conditions [74] and share of time in unfenced areas with other park users [21] were also raised.

## 4. Discussion

The results from Citespace analyses indicated that growing research focus has been directed on dog ownership, dog walking, and the physical environment since 2016. After the COVID-19 outbreak, growing numbers of researchers have emphasized the physical and mental health benefits brought by dog ownership and dog walking. Additionally, the ways to promote dog-related activities in urban settings have become a significant topic. In this context, dog parks or off-leash areas became important research foci, consistent with the direction of the body of research and illustrating their future research potential. 

As the outcomes in the Results section clearly illustrate the identified benefits, conflicts, and design/management strategies of dog parks (research questions a, b, c, d), the research question e: How is it possible to endorse the benefits and avoid the conflicts while determining the design and management strategies of dog parks still remains to be resolved. To explore the research question, we discussed the results of the systematic study from the perspectives of the following two questions.

### 4.1. Do the Existing Design/Management Strategies Address the Benefits and Conflicts in Dog Parks?

According to the results of the *PRISMA*, although most of the related studies explored both the pros and cons of dog parks and their design/management strategies, 9 out of 26 studies [26,44,53,55,60,62,71,74] developed strategies in response to the pros and cons of dog parks. Sixty-seven percent of these studies focused on hygiene issues, including dog waste, and solutions in dog parks [44,53,55,62,71,74]. Other studies, while discussing the pros and cons of dog parks and the associated importance of considering the benefits and conflicts for the construction and management of a dog park, did not explicitly discuss design and/or management strategies in terms of the benefits and conflicts brought by dog parks. For example, Lee et al. [22] investigated park user patterns, activities, and their satisfaction and perceptions to provide design guidelines for dog parks. Their findings aligned with the following research that dog parks contributed to the social and physical health of both park users and their dogs, and expanded the knowledge that the park design should be based on the evaluation of aspects of dog parks [22]. However, they overlooked the established connection between aspects of dog parks, especially the identified health benefits and limitations, and the design guidelines of an effective dog park. Both experimental and systematic studies started to propose design and management strategies for effective dog parks for enhancing their benefits while mitigating their risks [23,26,60]. Prior to 2020, a dog park design and management guideline considering both the advantages and disadvantages was generated based on the literature [21]. This design guideline was developed based on existing literature, which did not robustly examine the pros and cons aspects of dog parks. Additionally, the strategies have not clearly indicated which benefits they can bring and/or which issues they can mitigate, so the effectiveness of the design strategies is questionable. Existing research has developed design/management strategies for dog parks that address their benefits and conflicts, but how these strategies can effectively enhance these benefits and avoid the conflicts remains a significant research question to be explored.

### 4.2. How Can the Design/Management Strategies Endorse the Benefits and Avoid the Conflicts of Dog Parks?

Dog parks can bring users various benefits, but their improper design or management can lead to conflicts between dogs, their owners, other park users, and the physical environment. Based on the findings of the systematic study, we summarized the design and management strategies according to the frequency and relevance of the identified benefits, conflicts, and existing strategies (Table 2 and Table 3).

The identified articles inferred that a linear-based design could support both people and canine walking activities [3], which was aligned with the experimental evidence [77]. Significant numbers of the selected studies concluded that increased dog park access and proximity can encourage physical activities among dogs and their owners [21,22,30,33,39,46,49,59,64], because residents of local communities tended to walk to nearby dog parks more frequently. McCormack et al. [30] further discussed that a well-maintained dog park with clear signage could improve dog walking that contributes to physical health benefits. Recent statistical analysis indicated durable seating areas and adequate shade and shelter could facilitate social interaction among park users [78]. This reinforces the design strategies for investing in seating and shade trees to enhance social benefits in dog parks [32]. Physical and social health benefits among park visitors and their dogs are the most reported benefits brought by dog parks, but limited management strategies were developed for maximizing these benefits. It has been demonstrated that having events at parks, such as sports competitions, is correlated with physical activating and gathering of people [79], so we suggested that investing in events can encourage gathering visitors and dogs to engage in and physical and social activities in dog parks. Specific evidence between the research correlations in the dog park setting is anticipated to provide opportunities for future research.

As illustrated in Table 3, a greater number of design and management strategies were plotted to address conflicts and issues occurring in dog parks, when compared to enhancing their benefits. Hygienic issues in dog parks received the most attention from scholars. Dog feces, soil erosion, and damage to vegetation have received notable attention [53]. More concerning, health risks from disease transmission between dogs or from dogs to humans may occur without the proper design and management of gathering areas [80]. Regular monitoring programs and equipment, such as the placement of onsite surveillance cameras, can continuously supervise the condition of dog parks, including damage to vegetation and soil. Some scholars designed monitoring protocols with public engagement and mobile technology to examine hygiene issues and aggressive dogss [58,69]. However, surveillance programs should be carefully considered, as they can create issues of privacy. In addition to the park amenities, such as garbage cans, waste bags, and signage reminding and providing direction for waste disposal, the enhancement of the water system of a dog park is a key strategy. To minimize the transmission of zoonotic diseases, the location of dog parks should avoid proximity to natural water resources such as rivers and lakes [81], and Middle [26] suggested that seasonal drainage basin areas could be locations of choice for dog parks. Most importantly, management strategies corresponding to individuals and dogs can mitigate the spread of bacteria. The education of dog owners about environmental impacts to enhance self-enforcement of park users is the most effective strategy for decreasing the accumulation of dog waste and related hygienic issues in dog parks. This may be more critical than the waste-management amenities and strategies. In extreme cases, penalties and enforcement of a banned list of frequent offenders may also be necessary to mitigate these issues in dog parks.

Conflicts between park visitors with and without dogs for extended periods are especially prevalent in dog-gathering areas. Dogs that suffer from behavioral issues may trigger dog fights and aggregation-based dog conflicts, but also impact incompatibility between dog walkers and other park users. It is important for a well-designed and managed dog park to mitigate these issues. Among the listed strategies in Table 3, to mitigate conflicts, a logical park design with clear boundaries and proper fencing will separate dogs of different sizes and visitors with different intentions. Significant research including the most recent studies, indicates strengthening public engagement in the decision-making process is an effective solution to many controversaries [29,32,43,57,63,67,76]. Most existing issues in dog parks ultimately result from conflicts between different dogs, dog owners and non-dog owners, and impacts of ordinary dog park usage on environmental resources, such as the vegetation, soil, and other park uses. To relieve these conflicts, regular communication and cooperation between constituents, the local government, and stakeholders, including those who advocate for and oppose dog parks, are important in the public involvement process. The selection of dog park sites, design process, and daily management can all be enhanced through representation of different constituents. For example, the involvement of dog park activists and other residents in the process of determining a dog park’s location can resolve issues by taking into consideration the concerns of hygienic problem, noise and odors caused by the placement of a dog park from the beginning. Researchers can also be a vital part in the decision-making process by providing professional alternatives for relieving conflicts [29,57]. Additionally, an efficient negotiation mechanism will allow various members to mitigate dog park issues during the decision-making process. We concur with Toohey and Rock [62] that many problems created by the existence of dog parks should not be concealed but openly discussed. Various approaches, such as public meetings, anonymous emails, and online polls can work during the processes of the creation and use of a local dog park [23,82]. Routine evaluation of dog parks is also recommended for strengthening public engagement, enhancing their benefits, and alleviating their conflicts.

Although we provided the design and management strategies and distinguished the vital role researchers play in response to the identified benefits and conflicts of dog parks, some dilemmas in the existing research still need to be addressed, such as the lack of experimental evidence supporting specific aspects of dog parks and the strategies applied.

Growing numbers of studies have quantitatively explored associations between the features of built environment and dog-walking in Australia, Canada, and the USA. However, there is a dearth of experimental evidence about how the features/design of dog parks may influence park-based activities, such as dog walking. Arguments are passionate on both sides and debate has remained subjective and unresolved because experimental evidence of the ecological impacts of dog walking has been lacking. Holderness-Roddam [21] provided guidance for designing, planning, and managing dog parks primarily based on the literature. Some recent studies have begun to place focus on physical benefits and/or social components of dog parks. For example, Middle [26] proposed that increasing the accessibility of dog parks for neighborhood walking could bring a higher proportion of social interactions. Kresnye [69] designed a cooperative system addressing the physical and social experience of canines in dog parks. However, these strategies are primarily created through qualitative analysis, which not only lacks the establishment of reliability of rationale, but also challenges the generalizability of knowledge and quantitative comparison, such as a meta-analysis. Experimental evidence should be provided in future studies for the development of reliable design and management strategies for the progress of dog park development.

Physical health benefits among dogs and their owners going to dog parks are the most reported benefits in the systematic study. However, a national survey disclosed that dog walking was not sufficiently intensive so as to count as moderate to vigorous physical activity (MVPA) [83]. Evenson et al. [60] discovered people visiting a dog park tended to engage in sedentary activities, such as standing and watching dogs play, a finding supported by other two studies [22,84]. Recent studies have concluded that dog park users are less likely to engage in physical activities than other urban park users, which is contradictory to the previous self-reported results and would thus warrant further, more detailed investigation [26]. As for the differences between perceived physical benefits and the limitations on levels of physical activities, it is important for dog park design and management to support visitors who engage in various park-based physical activities. This leads to some suggestion that an effective dog park should increase general walkability and be accessible for potential dog walking residents. Consideration of the walkable surface with the degradation of grass in larger dog parks was proposed by Evenson et al. [60] to increase the levels of physical activity for dog park users. Park proximity and accessibility to a dog park is central to its health benefits, specifically through facilitating dog walking behaviors, which affirms previous findings by McCormack et al. [30] and Lee et al. [22]. Improving physical activity through dog walking is a promising public health strategy to improve health that could feasibly reach those who are sedentary [54]. Improving the routes to and from dog parks, such that owners can safely walk or jog with their dogs to and from the park, ultimately benefit people and canine physical health. Dog companionship provides social support for the owners to join group activities, and dog parks offer a destination for owners to go and join in activities with their dogs. In addition to dog walking, which was challenged as a sufficient MVPA, we advocate for the placement of exercise facilities, including human–canine specific exercise equipment, to facilitate dog owners to engage in intense MVPA with their dogs other than just walking. The organization of frequent activities and competitions can motivate the MVPA between dog owners and the dogs, which also contribute to the community engagement and social cohesion.

While some researchers elicited that the existence of dog parks in a community could improve the quality of life and the environment [21,22,60], and increase the property value [22], the controversies brought by off-leash areas may detract from the benefits of the dog park for a community. For such reasons, it is often controversial for municipal governments to plan for dog parks. The literature suggests to strengthen the connection to community-based dialogues for dog park planning and management. Graham and Glover [32] stated the social structure of dog park committees should be governed and managed by disadvantaged groups to increase the stewardship and communication with the community. Not only by strengthening the public engagement, especially the researchers’ involvement in the municipal governments’ decision-making process, but by attaching importance to the endogenous conflicts and public controversies caused by the canines as well, significant opportunities can be achieved to bring about positive changes to the relationships between urban residents and their canines [57].

## 5. Conclusions

Although dog ownership and dog walking bring various physical and social benefits, especially since the pandemic, dog parks, on one hand strengthen the benefits for people and their dogs; on the other hand, they cause contentious community issues because of the allowance and gathering of off-leash dogs in a public space. Hygienic issues and conflicts between dogs, park visitors with and without dogs are the most identified issues occurring in dog parks. Many people value the physical and social benefits of dog parks, but the objectively measured intensity of physical activity among dog park users is often lower than other park users.

Recent studies have started to develop design and management strategies for dog parks that address the benefits and conflicts. Our study advances these findings to specifically maximize the benefits and minimize drawbacks of off-leash areas. A number of corresponding strategies for the benefits and/or issues of dog parks are formulated based on the experimental evidence for urban parks, rather than specifically for a dog park setting. As there is a lack of empirical research exploring the associations between the design/management strategies and the benefits and conflicts of dog parks, there are research opportunities for experimental studies and greater sample sizes to fill the research gaps. Well-designed strategies for both the planning and management processes of dog parks can enhance the experience of dogs and their owners, while avoiding some of the conflicts that arise during visits to dog parks. The inevitable issues should be confronted and discussed through the decision-making process, from the placement and planning of a dog park to the daily management of the off-leash areas. This study contributes to an integrated understanding and the sustainable coexistence of canines, dog owners, and those human park users who do not own dogs in built environments through appropriate design and management strategies.

## Figures and Tables

**Figure 1 animals-12-02251-f001:**
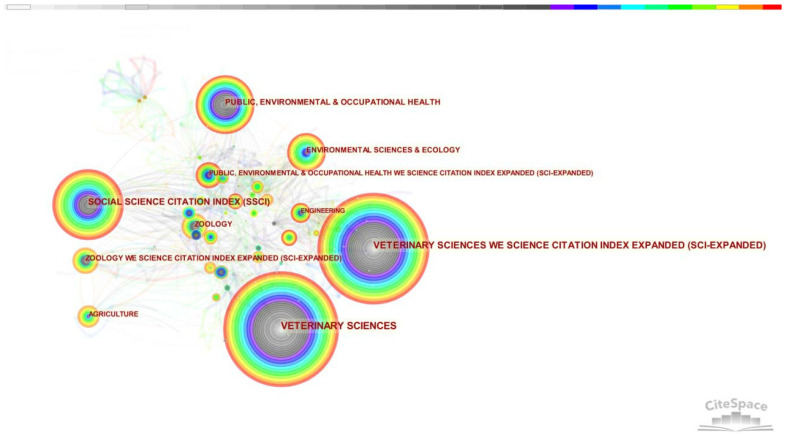
Disciplinary distribution structure of the research (one-year time slices with older data in “cooler” colors and newer data in “warmer” colors).

**Figure 2 animals-12-02251-f002:**
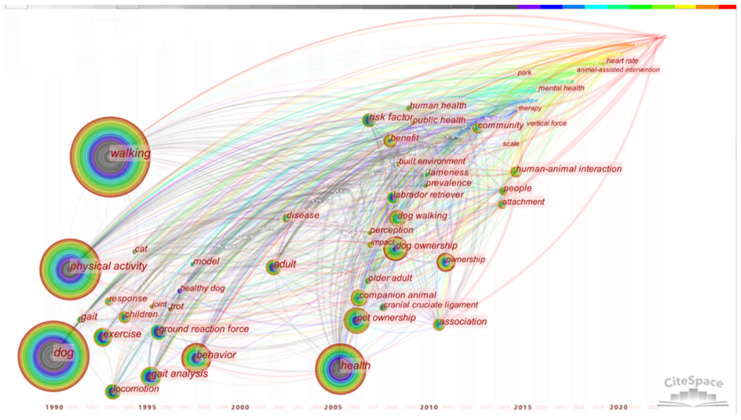
Time zone chart of keywords (each circle in the figure represents a keyword that first appears in the analyzed dataset and is fixed in the first year from the left side. If the keyword appears in a later year, it will be superimposed at the first occurrence).

**Figure 3 animals-12-02251-f003:**
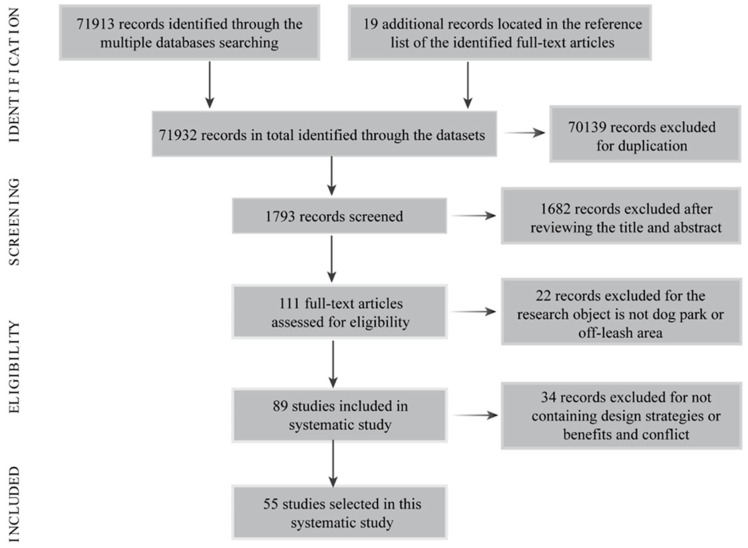
The flow of systematic study following the phases of PRISMA.

**Figure 4 animals-12-02251-f004:**
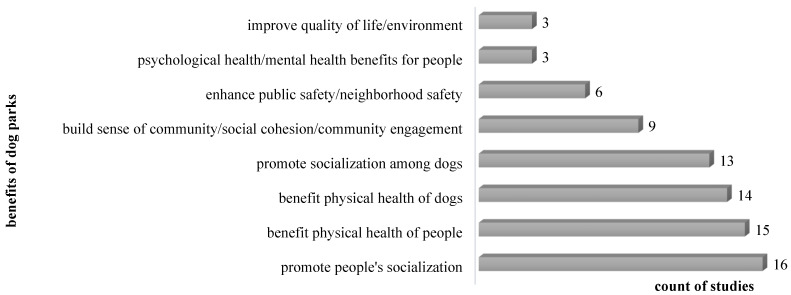
Count of studies identifying the benefits of dog parks.

**Figure 5 animals-12-02251-f005:**
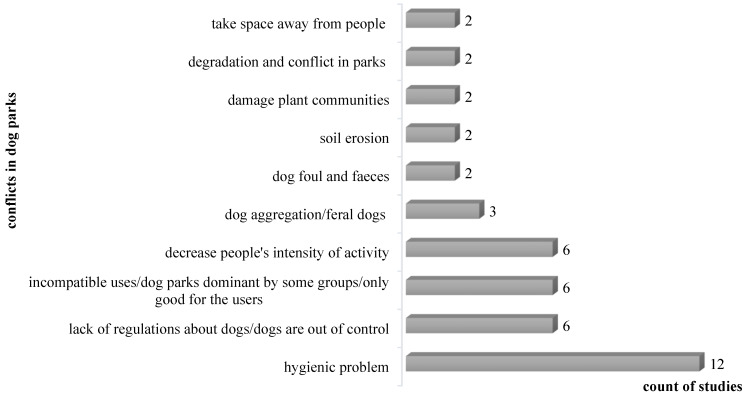
Count of studies identifying the conflicts occurring in dog parks.

**Figure 6 animals-12-02251-f006:**
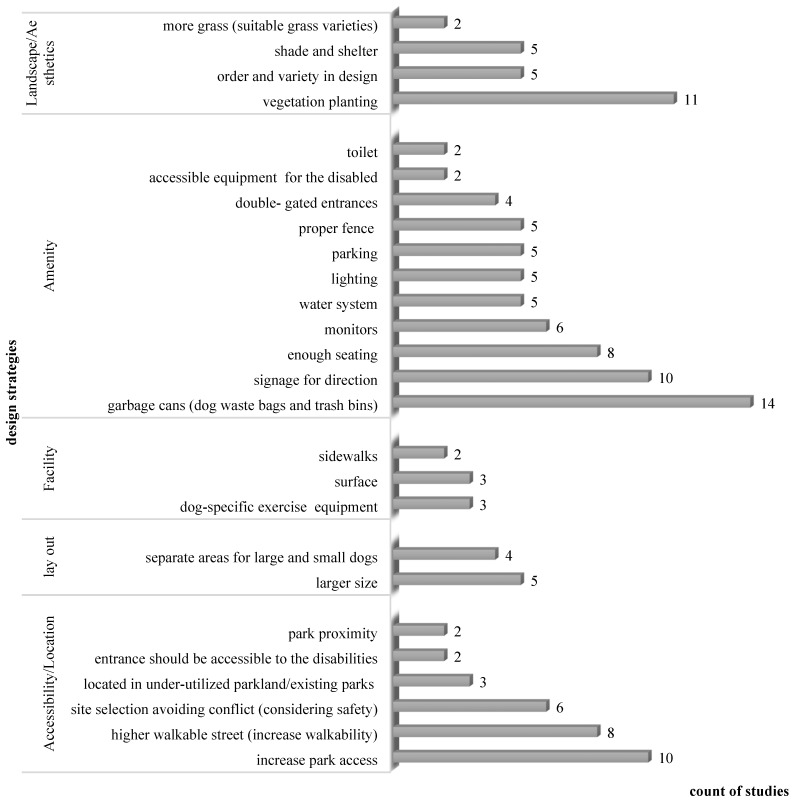
Count of studies summarizing the design strategies for dog park.

**Figure 7 animals-12-02251-f007:**
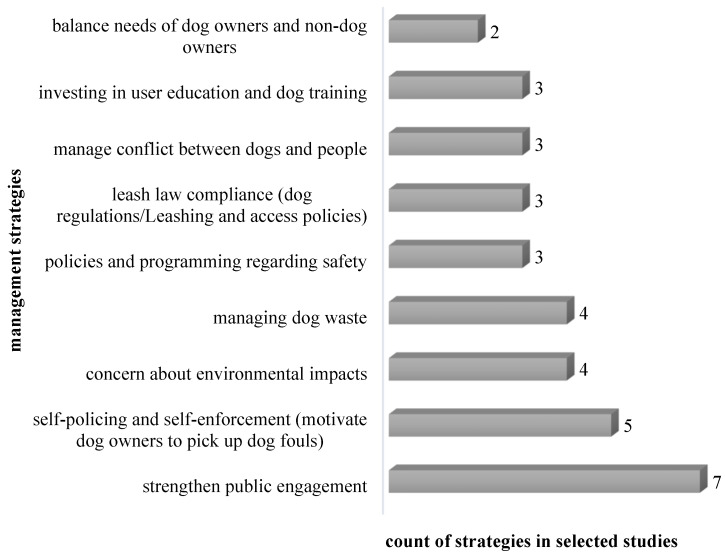
Count of strategies in studies summarizing the management strategies for dog parks.

**Table 1 animals-12-02251-t001:** Summary of the studies exploring pros and cons and/or the design/management strategies of dog park.

Articles	Benefits	Conflicts	Design Strategies					Management Strategies
			Access/Location	Lay out	Facility	Amenity	Landscape/Aesthetics	
Shyan et al., 2003 [47]		dog aggregation and fighting						
Forrest & Clair, 2006 [48]								leash law compliance
Allen, 2007 * [36]	promote socialization among dogs	inter-dog aggression		larger size		garbage cans	order and variety in design	animal control office
	benefit physical health of dogs			separate areas for large and small dogs		accessible equipment for the disabled	vegetation planting	penalty policy
						enough seating	shade and shelter	
Cutt et al., 2008 [49]			park access		dog-specific exercise equipment	signage		manage conflict between dogs and people
					surface options	garbage cans		
					proper fence	water system		
Lee et al., 2009 [22]	benefit physical health of dogs	decrease people’s intensity of activity	park access	separate areas for large and small dogs	dog-specific exercise equipment	water system	vegetation planting	invest in user education
	benefit physical health of people		site selection avoiding conflict	larger size	surface options	shade and shelter		invest in sponsoring events
	promote people’s socialization		connect with community trail system			enough seating		
	build sense of community		locations regarding safety			lighting		
	enhance public safety		double- gated entrance			parking		
	improve quality of life		accessible entrance for the disabled			signage		
	increase properties’ value							
Hazel et al., 2010 [50]					play equipment for children	lighting	more grass	self-policing and self-enforcement
					dog-specific exercise equipment	garbage cans	vegetation planting	
						enough seating	shade and shelter	
Iojă et al., 2011 [19]	benefit physical health of people	feral dogs						understand preferences of visitor categories
	benefit psychological health of people	hygiene problems						adapt parks to the size of flows
McCormack et al., 2011 [30]		decrease children’s intensity of activity	walkable street					
			park access					
Temple et al., 2011 [3]	benefit physical health of people				sidewalk		linear-based design	
Brown, 2012 * [39]	promote socialization among dogs		site selection avoiding conflict		dog-specific exercise equipment	double gate entrance	vegetation planting	
	benefit physical health of dogs		park access		surface options	water system	shade and shelter	
						proper fence		
						enough seating		
						signage		
						parking		
Throop et al., 2012 * [46]			locations regarding safety	separate areas for large and small dogs		parking	vegetation planting	blacklist
			park proximity			lighting		concern about environmental impacts
			double-gated entrance			proper fence		general maintenance
						water system		
						enough seating		
						toilet		
						garbage cans		
						signage		
Matisoff & Noonan, 2012 [34]				clear boundaries of users and resource				self-policing and self-enforcement
Jackson, 2012 [51]						monitors		self-policing and self-enforcement
Urbanik & Morgan, 2013 [24]	build sense of community	only good for the users						
	promote human’s behavior to dogs	take space away from people						
Toohey et al., 2013 [52]	benefit physical health of people		increase walkability					avoid conflict between dogs and other users
	build a sense of community							
Gómez, 2013 [29]	promote people’s socialization	take space away from people	site selection avoiding conflict					strengthen public engagement
	benefit physical health of people							
	enhance public safety							
	promote socialization among dogs							
	benefit physical health of dogs							
Lamotte, 2013 * [42]		hygienic problem						
Paradeis et al., 2013 [53]		hygienic problems					vegetation planting	fertilizer applications
		damaged plant communities					gardens and agriculture	technologies monitor soil
		soil erosion						
Richards et al., 2013 [54]	physical health benefits for people							
	mental health benefits for people							
	physical health benefits for dogs							
	mental health benefits for dogs							
Graham & Glover, 2014 [32]	contribute to social capital for the community				durable, low-maintenance seating	monitor		dog training and owner’s education events
	benefit dog socialization					lighting		strengthen public engagement
						garbage bins		managing dog waste
Gómez et al., 2014 * [41]	promote socialization among dogs							
	benefit physical health of dogs							
	promote people’s socialization							
	provide a safe place for dog to play off-leash							
Gaunet et al., 2014 [18]								dog regulations
Lowe et al., 2014 [55]		dog feces			path morphology	garbage bins		educate dog walkers about dog foul
Instone & Sweeney, 2014 [56]		dog waste						
Toohey & Rock, 2015 [57]	promote socialization among dogs	degradation and conflict in parks						strengthen public engagement
	promote people’s socialization	hygienic problem						face the conflicts
	vibrancy in the neighborhoods	dogs are out of control						
	benefit physical health of dogs	incompatible uses						
	benefit physical health of people	neighborhood problems						
Leung et al., 2015 [58]						monitors		
McCormack et al., 2016a [59]		decrease children’s intensity of activity	park access		dog-specific exercise equipment	garbage cans		policies and programming regarding safety
						signage		
Evenson et al., 2016 [60]	promote people’s socialization	limit physical activity		larger size	surface		vegetation planting	
	build a sense of community							
	improve quality of urban environment							
McCormack et al., 2016b [61]			increase walkability				aesthetical design	
							vegetation planting	
Rock et al., 2016 [62]		hygienic problems						policy focusing on dog-fouling
		dogs are out of control						
Engelberg et al., 2016 [13]			increase walkability				aesthetical design	
Burgess-Cady, 2016 * [40]	promote people’s socialization	cause degradation and conflict						
	benefit physical health of people	hygienic problems						
	promote socialization among dogs	dogs are out of control						
	benefit physical health of dogs							
Booth, 2017 [63]		damage plant communities						strengthen public engagement
		soil erosion						
		impacts on wildlife						
		incompatible uses						
Christian et al., 2017 [64]			strengthen street connectivity		sidewalks	signage	natural reserves	leashing and access policies
			park access			dog waste bags and trash bins		enforcement to preserve wildlife
			increase walkability			safety amenities		policies and programming on dog waste
								self-policing and self-enforcement
								less restrictions in public places
								balance needs of dog owners and non-dog owners
Christian et al., 2018 [65]	promote people’s socialization							
	benefit physical health of people							
	enhance public safety							
	promote socialization among dogs							
	benefit physical health of dogs							
Howse et al., 2018 [66]	promote socialization among dogs							
	benefit physical health of dogs							
Romo, 2018 * [44]		hygienic problems				garbage cans		
White et al., 2018 [20]	benefit physical health of people							
	promote people’s socialization							
Gómez et al., 2018 [67]	increase sense of community							strengthen public engagement
	promote social cohesion							policies and programming on safety issues
	increase neighborhood safety							policies and programming on dog waste
	promote people’s socialization							
Fletcher et al., 2018 [68]	promote people’s socialization	lack of regulations about dogs						
Veitch et al., 2019 [31]	benefit physical health of people	decrease children’s intensity of activity						
Kresnye et al., 2019 [69]						signage		
						monitors		
Gómez & Malega, 2020 [33]	benefit physical health of dogs		park proximity					
	promote socialization among dogs							
	promote people’s socialization							
Vincent, 2019 [70]	build social capital							
	benefit individuals’ health across the life span							
	strengthen community engagement							
Middle, 2020 [26]	promote people’s socialization	decrease people’s intensity of activity	located in under-utilized parkland	lager size		monitor	vegetation planting	
	enhance public safety	dog parks dominant by some groups	increase walkability					
Allen et al., 2020 [71]		hygienic problem				signage		
Koohsari et al., 2020 [1]	promote people’s socialization		street connectivity		sidewalks	enough seating		
	benefit physical health of people				dog-specific exercise equipment			
Holderness-Roddam, 2020 [21]	enhance public safety	hygienic problem	integrate dog parks into existing parks	separate areas for large and small dogs	surface options	proper fence	suitable grass varieties	time-share in unfenced area with other park users
	promote socialization among dogs	dogs are out of control	park access		dog-specific exercise equipment	signage	vegetation planting	policies and programming on safety
	benefit physical health of dogs		connect with community trail system	larger size		enough seating	shade and shelter	
	improve quality of urban environment		locate at least 150 ft from the residence			garbage cans		minimize environment impacts
	promote people’s socialization		accessible entrance for the disabled			water system		
	benefit physical health of people		double-gated entrance			toilet		
						lighting		
						parking		
Shealy, 2021 * [45]			increase walkability		surface	signage	esthetic green space	
						garbage cans	vegetation planting	
Westgarth et al., 2021 [72]	promote socialization among dogs		locations regarding safety			equipment for the disabled	avoid repetition scenery	
	benefit physical health of dogs		increase walkability			parking		
						enough seating		
						garbage cans		
Włodarczyk, 2021 [73]		hygienic problem						
		noise problem						
LaPointe, 2021 * [43]	attach strong emotion by dog walkers		integrate dog park into existing parks		garbage cans	monitors		minimize environment impact
								strengthen public engagement
Ebani et al., 2021 [74]		hygienic problem						periodical examinations
Scruggs et al., 2021 [75]								motivate dog owners to pick up dog fouls
								balance needs of pet owners and non-dog owners
Arnberger et al., 2022 [76]			site selection avoiding conflict	larger size				strengthen public engagement

* Academic dissertation/thesis.

**Table 2 animals-12-02251-t002:** Targeted benefits and their corresponding design and management strategies for dog parks.

Targeted Benefits	Corresponding Design Strategies	Corresponding Management Strategies
physical health benefits	increase walkability, park access and proximity; larger size of dog park; dog-specific exercise equipment; linear-based design; sidewalk	investing in events
social benefits	shade and shelter; sufficient seating	investing in events
safety enhancement	separate areas for large and small dogs; monitor; lighting; proper fence; double-gated entrances; signage for direction	strengthen public engagement; leash law compliance; self-policing and self-enforcement;policies and programming regarding safety
environment/ quality of life improvement	garbage cans; enhance water system; more grass (suitable grass varieties); order and variety in design	concern about environmental impacts; self-policing and self-enforcement; managing dog waste

**Table 3 animals-12-02251-t003:** Targeted conflicts and their corresponding design and management strategies for dog parks.

Targeted Conflicts	Corresponding Design Strategies	Corresponding Management Strategies
hygienic problem/dog fouling and feces	garbage cans and dog waste bags; enhance water system; signage; toilet; monitor	concern about environmental impacts; self-policing and self-enforcement; strengthen public engagement; managing dog waste; penalty policy; blacklist
dog aggregation/dogs are out of control	separate areas for large and small dogs; monitor; lighting; proper fence; double-gated entrances;	strengthen public engagement; investing in user education and dog training; self-policing and self-enforcement; animal control office; leash law compliance; blacklist
incompatible uses/dog parks dominant by some groups	site selection avoiding conflict (considering safety); locate at least 150 ft from the residence; clear boundaries for different users; signage; order in park design;	strengthen public engagement; balance needs of dog owners and non-dog owners; leash law compliance; time-share in unfenced area with other park user; self-policing and self-enforcement; blacklist
soil erosion/damaged planting and wildlife	more grass and suitable grass varieties; natural reserves	strengthen public engagement; periodical soil examination; fertilizer applications
Decrease people’s intensity of activity	increase walkability, park access and proximity; larger size of dog park; dog-specific exercise equipment; linear-based design; sidewalk	investment in events

## Data Availability

The data presented in this study are available on request from the corresponding author.

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
