# Peer review of "Benefits and Conflicts: A Systematic Review of Dog Park Design and Management Strategies"

_animals, 2022, doi:10.3390/ani12172251_

Round 1
Reviewer 1 Report
This manuscript presents an interesting study of “Benefits or Conflicts: A systematic study of dog park design and management strategies”. Overall, it combines a nice array of data using the PRISMA (Preferred Reporting of Items for Systematic Review and Meta-Analysis) methodology. This systematic study revealed the various type of advantages or disadvantages arising from dog parks design and management strategies. The paper clearly presents its objectives and research questions, thus it can make an important contribution to dog park designers or planners. In addition, this study synthesize and analyzes the aspects of dog park and assess if the strategies support the benefits while avoid the conflicts. Furthermore, the benefits and conflicts in dog parks are synthesized and sorted out according to the frequency of presence and the statistic results.
However, there are some aspects of the text that I believe should be addressed by the authors before publication. Firstly, the text needs extensive editing to correct typographical, grammatical and spelling mistakes. Examples are too many to be included in this report, but I could select a few just from the manuscript:
1. The key-words should be alphabetically arranged
2. Please use the reference style of this journal “Animals”
3. This sentence is too long, please we-write it, “To address these research gaps, this systematic study synthesizes and analyze the existing pro & cons and design/management strategies of dog parks and provide future planners, researchers, and decision makers recommendations for the optimization of design/research process of dog parks”.
4. Please correct the abbreviation of the term PRISMA as, “PRISMA (Preferred Reporting of Items for Systematic Review and Meta-Analysis)
5. I suggest this sentence may be referenced as,“Inclusion criteria of this study are those English-language peer-reviewed journal articles and academic dissertation/thesis while excluding the non-peer reviewed literature [Rashid et al 2020] which take dog parks as the theme and mentioned”.
Rashid, W., Shi, J., Rahim, I.u., Sultan, H., Dong, S., Ahmad, L., 2020. Research trends and management options in human-snow leopard conflict. Biological Conservation 242, 108413. https://doi.org/10.1016/j.biocon.2020.108413
6. Please use a uniform font size in all the tables. The Table 1 is having some words used in bold format.
7. Please correct the spelling of “design strategies” in the figure 6. Also, check font size that should be consistent with rest of the figures.
8. Please correct the sentence and re-write it “Most research suggested that strengthening the public engagement of the dog park decision-making process of the construction/management is a significant strategy addressing the conflicts between canine and human, also between dog owners and non-dog owners”.
9. This sentence requires correction, “Numerous researchers raised that the
environmental impacts caused by the canines should be attached more importance to the
management process.
1 . Please correct this sentence, “Specific evidence between the correlations in dog the park setting is prompted to provide in the future research”.
. This sentence needs correction, “Most exiting issues in dog parks ultimately result from the conflicts between different dogs, dog owners and non-dog owners, and between the environmental resources, such as the planting and soil and the gathering human and dog activities”.
. This sentence needs to be revised, “and we furthered the strategic findings to specifically maximize the benefits and minimize the issues of the off-leash area”.
. All the references should follow journal format for publication.
Author Response
We appreciate the reviewer’s thorough review and insightful comments throughout the manuscript. Our revised manuscript incorporated the reviewer comments and we found the revised manuscript is greatly improved as a result of these feedbacks. In the file, we have listed the reviewer comments in italics, and provided a point-to-point response/explanation of how we have addressed each critique. We apologized for the typographical, grammatical and spelling mistakes throughout the text. To improve the writing quality, an in-depth language editing was conducted by one of the native-speaker authors. Thank you very much for reviewing this work.

Reviewer 2 Report
This paper reviews the benefits and conflicts associated with dog parks in urbans areas and assesses whether design and management strategies emphasize benefits and minimize conflicts. The authors conclude that there is a lack of empirical research on issues related specifically to dog parks (most research focuses on urban parks generally). The review is thorough and current, but requires careful checking and editing for English word usage and grammar.
While the review is suitable for Animals, I am not sure that it is suitable for the Special Issue: Human-Wildlife Conflict and Coexistence in Urban Environments. The description of the SI includes feral dogs as wildlife, but owned dogs are the topic of this review. The only places where I found wildlife mentioned at all include the last sentence in the paragraph above Figure 5, the corresponding Booth (2017) citation in Table 1, and as a targeted conflict in Table 3; in these situations, it is the effects of off-leash owned dogs on wildlife.
Most of my other comments concern quality of presentation of information.
Title: I suggest changing “Benefits or Conflicts” to “Benefits and Conflicts.”
The paper lacks line numbers, author names, and affiliations.
Figures 1 and 2: The color and size of text in these figures makes the text difficult to read, and the font and style differ (e.g., all caps in Figure 1 and all lowercase in Figure 2); they should be consistent. Consider also increasing the size of both figures (Figure 3 is much easier to read). Finally, I do not understand the time zone aspect of Figure 2 (the text above this figure describes patterns before and after 2000, but where is 2000?), so perhaps more information is needed in the legend to ensure that readers fully understand what is being shown.
Last sentence above Figure 3: There was no Appendix with Table A in the pdf I downloaded. I suspect this should instead be Table 1, which is not called out in the text. Due to formatting of tables in Animals (e.g., all text centered within columns), Table 1 is very difficult to read and should be oriented or organized differently. Check with Animals staff or see how similar papers with such tables arrange information.
Figure 4 legend: Perhaps explain that the total number of studies is greater than 55 because some studies addressed multiple benefits?
Figure 5 legend: Perhaps explain that the total number of studies is greater than 55 because some studies addressed multiple conflicts?
Table 2: What do you mean by monitors (i.e., people onsite making sure things are going smoothly or surveillance cameras or something else)?
I hope the authors find my comments helpful.
Author Response
We appreciate the reviewer’s thorough review and insightful comments throughout the manuscript. Our revised manuscript incorporated the reviewer comments and we found the revised manuscript is greatly improved as a result of these feedbacks. In the attached file, we have listed the reviewer comments in italics, and provided a point-to-point response/explanation of how we have addressed each critique. To improve the writing quality, an in-depth language edition was conducted by one of the native-speaker authors to check and revise for the English word usage and grammar. Thank you very much for reviewing this work.

Reviewer 3 Report
Comments on the manuscript “Benefits or Conflicts: A systematic study of dog park design and management strategies” submitted to the Animals
General comments
I appreciate the opportunity to review this manuscript. This reviews dog park design, emphasizing the pros and cons. The topic is important to support policies for public spaces with dogs. The text is original, well-written, and sounds scientific. I have a few remarks about parts of the text not being understandable. Next, I will comment on some of these points.
Title
The title has an ambiguity about the topic addressed. It didn't seem to be about "Benefits or conflicts", a form of writing that excludes one of the terms in the approach throughout the text. It seems more appropriate to use the term "Benefits AND conflicts" because both approaches are presented in the text.
The word "study" is generic and does not make it clear to the reader that the manuscript is a systematic review, so I suggest changing the title.
Abstract
One of the most essential results found in the review is the lack of experimental studies on the pros and cons of dog parks. I suggest adding this result to the abstract.
Keywords
It is good.
Introduction
It is good
Methods
The inclusion of dissertations and theses presents some methodological problems. First, these textual productions are modified and published as articles in scientific journals. How did the authors ensure that there was no duplication of the same study in the form of a dissertation/thesis and scientific article? Second, there is much more production of dissertations and theses than articles. In many universities or research centers, the full text of theses and dissertations is embargoed and may be released only a few years later. In these cases, there is only the disclosure of the title, the evaluation committee, and the abstract. What were the inclusion and exclusion criteria for this type of textual production? Third, I miss focusing on what part of world studies is about. Is it in the EU? The USA? Canada? India? Considering that there are different spatial-cultural conceptions of parks (for dogs or not), this type of information provides a better understanding of the context for discussion. Finally, the evaluative rigor of articles (peer review) differs from theses and dissertations. Therefore, I do not consider that both types of textual production, articles, and theses/dissertations have the same weight in the type of analysis performed. Considering all these doubts, I strongly suggest removing theses and dissertations from the analysis.
“Additional literature that aligns with the search criteria, detected from the reference lists of the full-text articles were also included for subsequent screening”. Is any reference from Google Scholar, Scopus, PubMed, and Web of Science?
Please, check the "literatures" word along the text.
Results
Figure 2: You must explain what the colors represent, the size of the circles, and the lines scattered in the figure. Not every reader is familiar with the output of the analytical software used. This recommendation is for figure 1. While most of the terms exposed in the figure have to do with the approach and discussion of the manuscript, some terms raise doubts. For example, "cat", “cranial cruciate ligament”, “joint”, “ground reaction force” etc. How do understand these terms related to dog parks?
“The majority of these articles (26 out of 55) explored both the pros & cons and the strategies of dog parks…”. 26 out of 55 articles are 47% and in fact, it is not the majority.
Table 1 is important but in 7 pages within the main text, removing the reading flow. I suggest shifting the table and showing it as "Supplementary Material”.
In each figure from 4 to 7, the study count does not match the 55 selected articles. How to explain this discrepancy?
Discussion and Conclusion
It is good.
List do references
It is good.
Author Response
We appreciate the reviewer’s thorough review and insightful comments throughout the manuscript. Our revised manuscript incorporated the reviewer comments and we found the revised manuscript is greatly improved as a result of these feedbacks. In the attached file, we have listed the reviewer comments in italics, and provided a point-to-point response/explanation of how we have addressed each critique. Thank you very much for reviewing the work.

Round 2
Reviewer 3 Report
Dear authors
Thank you once again for the opportunity to review this manuscript. Substantial changes have occurred that have increased the quality of this manuscript. The new version has structural changes and a format that meets the criteria of coherence, understanding, clarity, and rigor for a scientific article. I'd just like to mention what appears to be an error in meaning on page 13: the term "feral dogs" refers to unowned, skittish dogs with no social contact or proximity to humans and generally occupying peri-urban spaces. I think in the sentence the authors mean "aggressive dogs", instead of "feral dogs".
With the modifications, the manuscript is more understandable and recommendable for publication, depending on the editorial decision.